# Mind, Mood and Microbiota—Gut–Brain Axis in Psychiatric Disorders

**DOI:** 10.3390/ijms25063340

**Published:** 2024-03-15

**Authors:** Corneliu Toader, Nicolaie Dobrin, Daniel Costea, Luca-Andrei Glavan, Razvan-Adrian Covache-Busuioc, David-Ioan Dumitrascu, Bogdan-Gabriel Bratu, Horia-Petre Costin, Alexandru Vlad Ciurea

**Affiliations:** 1Department of Neurosurgery, “Carol Davila” University of Medicine and Pharmacy, 050474 Bucharest, Romania; corneliu.toader@umfcd.ro (C.T.); razvan-adrian.covache-busuioc0720@stud.umfcd.ro (R.-A.C.-B.); david-ioan.dumitrascu0720@stud.umfcd.ro (D.-I.D.); bogdan.bratu@stud.umfcd.ro (B.-G.B.); horia-petre.costin0720@stud.umfcd.ro (H.-P.C.); prof.avciurea@gmail.com (A.V.C.); 2Department of Vascular Neurosurgery, National Institute of Neurology and Neurovascular Diseases, 077160 Bucharest, Romania; 3Neurosurgical Clinic, “Prof. Dr. N. Oblu” Emergency Clinical Hospital, 700309 Iași, Romania; dobrin_nicolaie@yahoo.com; 4Department of Neurosurgery, “Victor Babes” University of Medicine and Pharmacy, 300041 Timisoara, Romania; 5Neurosurgery Department, Sanador Clinical Hospital, 010991 Bucharest, Romania

**Keywords:** gut microbiota, probiotics, antipsychotics, antidepressants, MDD, autism, brain–gut axis

## Abstract

Psychiatric disorders represent a primary source of disability worldwide, manifesting as disturbances in individuals’ cognitive processes, emotional regulation, and behavioral patterns. In the quest to discover novel therapies and expand the boundaries of neuropharmacology, studies from the field have highlighted the gut microbiota’s role in modulating these disorders. These alterations may influence the brain’s processes through the brain–gut axis, a multifaceted bidirectional system that establishes a connection between the enteric and central nervous systems. Thus, probiotic and prebiotic supplements that are meant to influence overall gut health may play an insightful role in alleviating psychiatric symptoms, such as the cognitive templates of major depressive disorder, anxiety, or schizophrenia. Moreover, the administration of psychotropic drugs has been revealed to induce specific changes in a microbiome’s diversity, suggesting their potential utility in combating bacterial infections. This review emphasizes the intricate correlations between psychiatric disorders and the gut microbiota, mentioning the promising approaches in regard to the modulation of probiotic and prebiotic treatments, as well as the antimicrobial effects of psychotropic medication.

## 1. Introduction

Neuropharmacology is predicated on the foundational concept that pharmacological agents concurrently exert multifaceted impacts across various regions of the brain and body, thereby eliciting a diverse array of functional and behavioral responses during both somnolent and wakeful states. The discipline’s structural framework is intrinsically interdisciplinary, drawing upon a spectrum of scientific domains. Integral contributions stem from cellular biology, molecular biology, and biochemistry, along with various sub-disciplines of neuroscience, ranging from neuroanatomy to neurophysiology. These fields have historically provided, and persist in providing, critical insights that underpin the advancement of neuropharmacological research [1,2,3].

A mental disorder is typified by a clinically notable perturbation in an individual’s cognitive processes, emotional regulation, or behavioral patterns. This condition is commonly correlated with a degree of distress or a decrement in proficiency in crucial functional domains [4]. The heterogeneity of mental disorders underscores the imperative to delineate efficacious therapeutic modalities for psychiatric conditions, a challenge of significant relevance in contemporary societies. The impasse encountered in the discovery of novel pharmacological agents has prompted a re-examination of previously studied compounds [5,6,7]. Additionally, it has been documented that neuropsychiatric disorders, along with the themes of brain connectivity and emotion, constitute some of the most frequently cited topics within the realm of neuroscience, as evidenced by the literature [8].

In the classical curriculum of pharmacokinetics and pharmacodynamics, the influence of the gastrointestinal (GI) tract’s microbiota on the disposition and action of drugs is often underemphasized. However, it is increasingly recognized that the pharmacological fate and efficacy of drugs are determined not solely by the host itself but also significantly by the complex consortium of microorganisms residing in the GI tract, collectively referred to as the gut microbiota [9]. The intestinal flora comprises approximately 10^14^ microbial cells, predominantly characterized by two principal bacterial phyla: *Bacteroidetes* and *Firmicutes*. These microorganisms collectively represent 75–80% of the total gut microbiome, thereby constituting a significant majority of the intestinal microbial ecosystem [10]. The phyla *Proteobacteria*, *Actinobacteria*, *Fusobacteria*, and *Verrucomicrobia*, while constituting a relatively smaller fraction of the gut microbiome, engage in critical interactions with other intestinal microorganisms. These interactions play a pivotal role in human health, particularly in modulating the functions of the immune and nervous systems. This interplay underscores the integral role of these microbial groups in the broader context of human physiological processes [11]. In a state of physiological well-being, these microbial populations maintain a symbiotic equilibrium with the activity of host cells, thereby facilitating the homeostatic regulation of the gut–brain axis. This balance is essential for optimal functioning of interconnected biological systems [12] (Figure 1). Currently, the scientific community is increasingly focusing on the utilization of natural supplements aimed at augmenting the intestinal microbiome, with a consequent beneficial impact on cerebral functions. Among these, the employment of pro- and prebiotic dietary supplements has garnered notable popularity. These supplements are recognized for their positive influence on the intestinal microbiome, thereby enhancing both gastrointestinal health and the functionality of the gut–brain axis. This enhancement holds promise in regard to playing a significant role in the prevention and treatment of certain mental disorders, underlining their potential therapeutic applications [13].

In the complex realm of human cognition and physiology, the interactions between mind, mood and the brain–gut microbiota arises as an intriguing frontier. This implies that the gut microbiota evolves alongside the gut–brain, brain and, consequently, the mind. Moreover, the microbiota play an insightful part in both the maturation and operation of the brain, thus suggesting their influence in the complex orchestra that regulates our mood [10,11]. Major Depressive Disorder (MDD), Anxiety, and Bipolar Disorder are psychopathologies primarily affecting the mind, hence categorized as mind disorders, influenced by alterations of the BGM.

It is of utmost importance to comprehend the vast multicontextual interactions and reactions of the molecular structures that govern over our physiology in a way which will lead us to discover new avenues of treating different psychiatric disorders. Perhaps by analyzing the pathways that form the gut–brain axis and their mechanisms, we may one day have the ability to address the root of psychiatric diseases, a root which still lays somewhere deep, buried under yet to be discovered mechanisms. This review sheds light on different insights regarding the gut–brain microbiota’s influence on psychiatric disorders, including discussions of prebiotic and probiotic-based therapies that have succeeded in managing psychopathologies, thus setting the context of the complex relationship between molecules and these ruthless conditions.

## 2. Brain and Intestinal Microbiotic Interaction: A Molecular Perspective

Introduction to the brain–gut axis

The gut–brain axis (GBA) can be characterized as a multifaceted bidirectional communication system that establishes a connection between the enteric and central nervous systems [14]. This system transcends mere structural linkage, extending its functionality through endocrine, humoral, metabolic, and immune-mediated signaling pathways. Critical components constituting this axis include the autonomic nervous system, the hypothalamic−pituitary−adrenal (HPA) axis, and the neural network within the GI tract. Such components enable an interactive relationship between the gut and the brain, thereby allowing the brain to exert regulatory influences on various intestinal functions, inclusive of the modulation of functional immune effector cells in the GI tract. In a reciprocal manner, the gut imparts significant effects on neurological aspects, namely mood, cognition, and mental health [15].

Communication within this system is structured across four discrete levels: Neuroanatomical, neuroendocrine, immunological, and a level comprising neurotransmitters, neuropeptides, and microbiome-derived products. Each level represents a unique facet of interaction, contributing to the overall functionality of the system [16]. Perturbation of the aforementioned levels within the brain–gut axis is implicated in the emergence of diverse disorders, as illustrated in Figure 2. Within the neuroanatomical dimension, two distinct pathways operate. The primary, direct conduit linking the GI tract and the brain involves the vagus nerve and the autonomic nervous system (ANS). In contrast, the secondary, indirect route encompasses the connectivity between the enteric nervous system (ENS) and the ANS [16]. These neuroanatomical pathways are critically involved in various psychiatric disorders. Chronic stimulation of the vagus nerve (VNS) produces an anxiolytic effect by enhancing excitatory neurotransmission [17], mediated by the alpha-amino-3-hydroxy-5-methyl-4-isoxazole propionic acid receptor (AMPAR). Concurrently, transcutaneous auricular VNS has been demonstrated as an efficacious and safe technique for reducing depression scores, showing comparable effectiveness to traditional antidepressants [18]. Elevating vagal tone has been associated with a reduction in cytokine production, indicating an immunomodulatory effect [19]. The gut microbiota (GM) exhibits anti-stress and anti-anxiety properties by modulating the activity of the vagus nerve and secreting neurotransmitters, including γ-aminobutyric acid (GABA), serotonin, and short-chain fatty acids. These mechanisms and their implications will be further elaborated on in the subsequent chapter [20,21].

At the neuroendocrine level, a primary component is the stress-responsive HPA axis. This axis originates from the paraventricular nucleus of the hypothalamus, which synthesizes corticotropin-releasing hormones (CRHs). The CRHs, in turn, stimulates the anterior pituitary gland to release adrenocorticotropic hormones (ACTHs). The ACTHs subsequently prompt the adrenal glands to secrete glucocorticoids [16]. The gut microbiota modulates nutrient availability, thereby impacting the secretion of biologically active peptides from enteroendocrine cells. This alteration in peptide release can subsequently influence the functioning of the gut–brain axis [22]. Therefore, the neuropeptide galanin enhances the activity of the central component of the HPA axis, influencing the secretion of corticotropin-releasing factors (CRFs) and ACTHs. This stimulation subsequently leads to an increased secretion of glucocorticoids from the adrenal cortex [23]. Furthermore, a study conducted by Flores-Burgess et al. introduces the potential of utilizing a combination of galanin (1–15) and fluoxetine as an innovative approach for treating depression. This suggests the presence of underlying immune and neuroregulatory mechanisms that may mediate the impact of stress on the gastrointestinal tract [24]. Cortisol exerts its influence within the central nervous system (CNS) through both hormonal and neural communication pathways. These interacting pathways collectively affect the activities of various cell types within the gut, including intestinal effector cells, smooth muscle cells, epithelial cells, enterochromaffin cells, interstitial cells of Cajal, enteric neurons, and immune cells [25]. Consequently, conditions of stress lead to alterations in the gut microbiome, immune function, mucus production, intestinal motility, and permeability [26].

Regarding neurotransmitters, histamine, which is produced by gut microbes, has been identified as having a correlation with the GBA, exhibiting immunomodulatory and anti-inflammatory properties [27]. The mechanisms linking histamine to depression involve the stimulation of mast cells by kynurenine metabolites. Additionally, the activation of the indoleamine 2,3-dioxygenase (IDO) enzyme by histamine, which catalyzes the conversion of tryptophan to kynurenine, contributes to depression through the accumulation of kynurenine [28]. The ratio of kynurenine to tryptophan (K/T) demonstrates a significant correlation with both the onset and severity of depressive symptoms. This correlation is particularly evident following the administration of IDO inducers like interferon-β or exposure to stress [29,30]. Consequently, stress-responsive cells that produce histamine, particularly mast cells, are identified as potential therapeutic targets in the treatment of depression [31].

In the context of immune pathways, cytokines serve as key mediators. These cytokines enter the bloodstream and are conveyed to the brain via the GBA [32]. For an extended period, the role of the immune system in major depressive disorder (MDD) has been considered significant. For instance, depressive symptoms frequently accompany infections in humans, and individuals with autoimmune disorders often experience comorbid depression [33]. Inflammation can lead to the disruption of the blood–brain barrier, induce cellular and structural alterations in the central nervous system, prompt the release of glutamate from microglia, and impair long-term potentiation [34]. Glial cells, encompassing microglia, astrocytes, oligodendrocytes, and ependymal cells, engage in interactions with neurons. These interactions play a crucial role in influencing brain health and can contribute to the development of MDD [35]. Glial functions can be influenced by the gut microbiota through neural and chemical signaling pathways. The gut microbiota plays a pivotal role in modulating microglial activation states, ranging from pro-inflammatory to anti-inflammatory. Dysfunctional microglia can initiate signaling cascades that lead to neuroinflammation, which is a key factor in depression [35,36]. For instance, rifaximin and minocycline have been shown to be effective in mitigating stress-induced depressive-like phenotypes. The antidepressant-like effects of these substances are linked with changes in microbial composition and metabolites. These changes subsequently lead to alterations in brain functions, brain microglia (Iba1), and peripheral inflammatory cytokines, including the tumor necrosis factor (TNF)-α, interleukin (IL)-1β, IFN-γ, and IL-12 [37,38].

The interaction between microbiota and the immune system is significantly mediated by metabolites derived from microbes, such as short-chain fatty acids (SCFAs) and tryptophan metabolites. These metabolites also play a crucial role in the context of psychopathologies, a topic that will be further elaborated upon [39].

Microbiota Imbalance: A Gateway to Psychopathologies

GM consists of a diverse array of microorganisms residing in the GI tract, including archaea, bacteria, protists, and fungi [40]. Research indicates that the genetic diversity of the GM surpasses that of the human body by approximately 150 times. It comprises about 100 billion bacteria, encompassing roughly 1000 species and approximately three million genes [41]. Historically, eminent physicians like Hippocrates and Galen acknowledged the fundamental role of digestive processes in health, famously asserting that “maldigestion is the root of all suffering” and emphasizing that “all diseases originate in the intestine”. They utilized the available treatments of their era, such as herbal infusions, to alleviate symptoms of various diseases [42]. Individual variations in the GM significantly influence human growth, dietary needs, physiological changes, and genetic differences. It has been observed that the composition of the GM varies according to factors such as age, gender, geographic location, dietary habits, and genetic variations [43]. The GM exert an influence on brain function during various cellular processes, including axonal processing, apoptosis, myelination, synaptogenesis, and cell differentiation. They also impact cognition during neurogenesis, particularly in the presence of growth hormones such as circulating IGF-1 [44,45].

The intestinal microbiota play a significant role in the modulation of neuroregulatory substance secretion from the CNS, including neurotransmitters like dopamine, serotonin, and melatonin. These substances are deeply involved in regulating mood and mental functions. Consequently, the microbiota can be a crucial factor in the management of mood disorders, such as depression and anxiety [15,46].

MDD, a neuropsychiatric condition, is characterized by features of immune dysregulation [47]. The gut microbiome synthesizes various molecules that can impact brain function, including neurotransmitters (biogenic amines, acetylcholine, and GABA, SCFAs, indoles (tryptophan and its metabolites), bile acids, choline and its metabolites, lactate, and vitamins [48]. It is posited that these molecules may influence depressive behavior through multiple pathways: direct stimulation of brain receptors; activation of neural, endocrine, and immune mediators in the periphery; and epigenetic mechanisms, including histone acetylation and DNA methylation [48].

Neurotransmitters that are produced, either directly or indirectly, by gut bacteria have the potential to influence emotional behavior. This influence is exerted through the binding of specific receptors located in the CNS, or on peripheral receptors found on neural or immune cells [48]. A research study that involved the chronic administration of L. rhamnosus to mice observed alterations in the expression of GABAA and GABAB receptors along with changes in brain activity levels. This was accompanied by a noticeable reduction in anxiety and depression-like symptoms in the mice [49]. While augmenting serotonin (5-HT) production in the gut does not lead to increased central concentrations of 5-HT [50], the central levels of 5-HT can be elevated by boosting the concentrations of its precursor, tryptophan, in the GI tract [51].

In a multitude of studies, it has been observed that derivatives of tryptophan, tyrosine, and purines exhibit differential expression in patients with MDD. This suggests that the kynurenine pathway’s metabolic components might play significant roles in the pathophysiology of the disease [52,53,54]. Furthermore, distinct microbial compositions have been identified in these patients compared to healthy individuals. Notably, individuals with depressive disorder exhibit elevated levels of *Actinomycetota*, *Pseudomonadota*, and *Bacteroidota phyla*, alongside diminished populations of the *Bacillota* and *Lactobacillaceae phyla* [25]. Chen et al.’s investigation into GM dysbiosis in women with MDD revealed significant enrichment of *Bacteroidetes*, *Proteobacteria*, and *Fusobacteria* phyla in patients, whereas higher levels of *Firmicutes* and *Actinobacteria* phyla were observed in healthy controls [55]. Species from the *Lactobacillaceae* family have been identified as possessing antidepressant and anti-inflammatory properties. Additionally, an increased presence of Enterobacteriaceae and *Allistipes* genera bacteria, coupled with reduced prevalence of the *Faecalibacterium* genus, correlates with the severity of depressive symptoms [56,57,58].

SCFAs are small organic molecules generated in the cecum and colon through anaerobic fermentation of predominantly non-digestible dietary carbohydrates. These compounds not only cross-feed other bacteria but are also efficiently absorbed in the large bowel [59]. SCFAs play critical roles in digestive, immune, and central nervous system functions. There is a variety of evidence regarding their impact on behavior; for instance, administering the three most abundant SCFAs—acetate, butyrate, and propionate—has been shown to reduce symptoms of depression in mice [60]. In a study by Yu et al., the differences in GM and SCFAs in the serum of patients with first-episode depression compared to a healthy population were examined. It was found that the MDD group had lower levels of certain gut microbiota, including *Akkermansia*, *Megamonas*, the *Prevotellaceae* NK3B31 group, butyrate-producing bacteria like *Lachnospira*, *Subdoligranulum*, *Blautia*, *Dialister*, and acetate-producing *Streptococcus*. This suggests that changes in these intestinal bacteria may be involved in the pathogenesis of MDD [61]. Additionally, SCFAs, particularly butyrate, acetate, and propionate, are known to possess histone deacetylase (HDAC) inhibitory activity, which leads to transcriptional activation by promoting euchromatin configuration, highlighting their role in epigenetic regulation [21].

The gut microbiota Is also implicated in the development of mood disorders through its association with a weakened intestinal barrier, a characteristic of intestinal dysbiosis [50]. This compromised barrier often correlates with increased expression of pro-inflammatory cytokines in individuals exhibiting depressive symptoms. These cytokines include IL-1β, IL-6, TNF-α, interferon gamma, and elevated levels of C-reactive protein [57,62]. The gut microbiota is known to influence the transcription of these cytokines, with dysbiosis activating what is termed the inflammatory pathway. Conversely, beneficial metabolites like SCFAs are observed to restrict the production of pro-inflammatory cytokines, such as IL-1 [63].

In the context of treatment, anti-inflammatory drugs, specifically COX-2 inhibitors, have demonstrated effectiveness in treating major depression. For instance, Müller et al. conducted a study on the therapeutic potential of the COX-2 inhibitor celecoxib in treating MDD. In their study, 40 patients experiencing acute depressive episodes were divided into two groups, both of which showed significant improvement in depression scores following treatment [64].

Bacterial lipopolysaccharides (LPSs), predominantly originating from Enterobacter spp., have been identified as playing a significant role in major depression. Notably, LPS levels are higher in patients with severe depression compared to healthy individuals [65]. These metabolites can enter the systemic circulation through permeability defects in the intestinal tight epithelial junctions, leading to leaky gut syndrome. This condition subsequently prompts the production of antibodies against LPS, potentially further destabilizing the gut–brain microbiota axis [66]. Van Eeden et al.’s research demonstrated a strong association between basal and LPS-stimulated inflammatory markers and MDD symptoms in their study participants. This suggests that anti-inflammatory strategies could be beneficial for certain subgroups of MDD patients [67].

In animal models, specifically mice, intraperitoneal injection of LPS induced depressive-like behavior. Subsequent treatment with sodium butyrate alleviated these changes, highlighting the detrimental impact of translocated bacteria and LPS, as well as the beneficial role of butyrate in the pathophysiology of depression [68].

Anxiety disorders, a group of mental illnesses characterized by heightened sensitivity to stressful stimuli (anxiety, fear, or panic) without a clear justification for these feelings, encompass conditions such as generalized anxiety disorder, panic attacks, post-traumatic stress disorder (PTSD), and various phobias (e.g., agoraphobia) [69]. Recent research has increasingly focused on the link between the pathophysiology of “stress” and the gastrointestinal tract, thereby implicating gut microbiota in this relationship [70,71]. Stressful environmental and psychosocial factors significantly impact gastrointestinal function and the immune system, consequently affecting the microbiota in either short-term or prolonged/chronic manners [56]. Episodes of intense anxiety or fear can lead to digestive issues such as indigestion or gastric pain, while long-term stress associated with anxiety disorders is implicated in severe gastrointestinal disorders like esophageal reflux, gastric ulcers, and inflammatory bowel syndromes (e.g., ulcerative colitis) [72].

Stress-induced effects on the gastrointestinal tract via the intestinal−brain axis notably include alterations in intestinal motility, visceral discomfort, changes in secretory capacity, and intestinal mucosal permeability [73]. These alterations significantly influence the composition and functionality of the gut microbiota, potentially leading to dysbiosis, which in turn may exacerbate anxiety symptoms within the gut/brain axis framework [74].

Generalized Anxiety Disorder (GAD) shares many characteristics with MDD, often complicating differential diagnosis. A study by Dong et al. utilized 16S rRNA gene-sequencing analysis to compare gut microbiome compositions, revealing distinct microbial profiles in GAD. Specifically, the abundances of the *Fusicatenibacter* and *Christensenellaceae_R7_group* were significantly reduced in GAD compared to healthy individuals, while *Sutterella* was more abundant and *Faecalibacterium* less so in GAD relative to MDD [75]. Additionally, Butler and colleagues conducted whole-genome shotgun analyses of 49 fecal samples to investigate compositional and functional differences in the gut microbiota of patients with Social Anxiety Disorder (SAD) versus healthy controls. Their findings highlighted an increased presence of *Anaeromassillibacillus* and *Gordonibacter* genera in SAD patients, while *Parasuterella* was more prevalent in healthy individuals [76].

There has been significant progress in addressing anxiety disorders through various approaches. One such development is the use of the probiotic strain *Lactobacillus plantarum P-8*, which has shown promising results in treating stress and anxiety in adults [77]. Additionally, proof that mindfulness-based cognitive therapy is effective in populations with high trait anxiety provides further evidence of gut–brain communication [78]. Furthermore, a *L. plantarum JYLP-326* intervention has been found to potentially alleviate anxiety, depression, and insomnia in test-anxious college students. The mechanism behind this effect is thought to be related to the regulation of gut microbiota and fecal metabolites [79].

Research into GM has extended to exploring its role in neuropsychiatric disorders such as Autism Spectrum Disorder (ASD). Notably, approximately 40% of individuals with ASD experience gastrointestinal problems [80,81]. Studies have shown that GM can influence mood and behavioral changes from childhood to adulthood [82]. The microbiome, which colonizes the gut immediately after birth, connects to the brain as the child grows. Any inflammation or impediment during this development process can result in impaired cognition, mood and memory changes, and atypical behavior [83]. Epidemiologically established risk factors such as maternal exposure to the anticonvulsant valproate, maternal inflammation during pregnancy, and maternal obesity have been found to alter GM composition in ASD animal models [84,85]. ASD has been linked to GM species that are vulnerable to vancomycin and contribute to a pro-inflammatory condition [86]. Probiotics (beneficial live microbial cultures) and/or prebiotics (beneficial non-digestible carbohydrates like fibers) have shown an ability to modulate social behavior in animal studies [87]. These findings are particularly intriguing as they may be applicable to humans, potentially leading to novel microbiota-based therapies for ASD treatment.

Thus, considering what was stated until now, we can assume the existence of a profound connection between the brain and the gut, with an emphasis on the GM dysbiosis and its effects, as is illustrated in Figure 3.

Clinical evidence increasingly supports the notion that the enteric microbiota has a substantial and profound impact on the gut–brain axis, influencing aspects such as mental state, emotional regulation, neuromuscular function, and regulation of the HPA axis. A study by Schaub et al. highlights that augmenting standard care with probiotic treatment can enhance outcomes in depressive symptoms, alongside observable changes in gut microbiota and brain function. This finding bolsters the understanding of the microbiota-gut–brain axis in MDD. Notably, the study demonstrated that the group receiving probiotics exhibited a more significant reduction in depressive symptoms compared to the placebo group, suggesting the efficacy of probiotics in enriching specific microbial taxa [88]. Further exploration of the relationship between psychopathologies and the gut microbiota is detailed in various studies, as exemplified in Table 1 of the referenced publication.

Studies have increasingly clarified the mechanisms through which microbiota directly and indirectly influence the emotional and cognitive centers of the brain [14], with it being shown that variations in microbiota composition are linked to changes in these neural communication systems [93]. Given these findings, there is a pressing need for specific treatments targeting conditions associated with gut microbiota alterations.

Among the therapeutic approaches being explored, interventions such as probiotics, prebiotics, symbiotics, and fecal microbiota transplantation are under investigation for their potential in treating neurological disorders derived from gut microbiota imbalances. These therapies are part of a growing field of research, offering promising avenues for addressing these complex conditions. The exploration of these treatments and their mechanisms will be further detailed in the subsequent chapter, providing a comprehensive overview of the current state of knowledge and potential future directions in this field [94,95].

## 3. Prebiotics and Probiotics in the Treatment of Psychopathologies

Probiotics are live microorganisms that, when administered in adequate amounts, confer health benefits to the host by colonizing the body. They are capable of adjusting the structure of human intestinal microorganisms and inhibiting the colonization of pathogenic bacteria in the intestine [96]. Additionally, probiotics are known to aid in the development of a healthy intestinal mucosal layer, thereby enhancing the intestinal barrier function and improving immunity [97,98]. Understanding the mechanisms through which probiotics act on the human body, as well as promoting their growth and reproduction, is essential. This is closely linked to the role of prebiotics.

Prebiotics are components, primarily polysaccharides, that the human body cannot digest or absorb. They contribute to the growth or activity of beneficial microorganisms within the host [99]. These substances are known for their roles in enhancing immune regulation, combating pathogens, impacting metabolism, increasing mineral absorption, and overall health promotion [100]. Prebiotics often include certain polysaccharides, oligosaccharides, microalgae, and natural plant sources, and they are widely available [101]. Studies, particularly in rodents, have shown that neurobiological processes related to cognition and affect are influenced by the gut microbiota. Certain dietary fibers capable of altering the gut microbiota composition are thus categorized as prebiotics [102]. Emerging prebiotic sources include algae, fruit juices, peels, seeds, traditional Chinese medicine, and microorganisms involving polysaccharides, polyphenols, and polypeptide polymers [103]. Gibson et al. have outlined the key characteristics of prebiotics, which include fermentation by intestinal microflora, resistance to digestion by mammalian and bacterial enzymes, resistance to gastric acidity, the ability to withstand food processing, and selective stimulation of the growth and activity of beneficial intestinal bacteria such as *Lactobacilli* or *Bifidobacteria* [104].

Numerous studies have provided evidence of the positive effects of probiotic and prebiotic therapies on mental mood and psychopathological diseases, as detailed in Table 2 of the relevant publications. The beneficial impact of probiotics on anxiety disorders underscores the significant role of microbiota in their development [105]. Specifically, the probiotic Bifidobacterium longum and Bifidobacterium infantis have shown efficacy in alleviating depression and anxiety symptoms associated with Irritable Bowel Syndrome (IBS), potentially through an increase in 5-hydroxytryptophan levels derived from tryptophan [106,107].

A four-week intervention study demonstrated that a galactooligosaccharides (GOS) prebiotic supplement could effectively improve pre-clinical anxiety indices [108]. Additionally, the anxiety-reducing effects of probiotics have been significantly documented in populations with anxiety disorders [109]. A meta-analysis revealed that both probiotics and synbiotics significantly reduced anxiety scores [110]. In a randomized controlled trial by Eskandarzadeh et al., a combination of probiotics and sertraline was superior to sertraline alone in reducing anxiety symptoms over 8 weeks in patients with Generalized Anxiety Disorder (GAD), suggesting an effectiveness in combined therapies [111].

Probiotics impact anxiety improvement through several mechanisms, such as promoting the ENS or stimulating the immune system through bacteria. They can decrease systemic inflammation and regulate the hypothalamic–pituitary–adrenal axis stress response [112]. Additionally, these substances induce the secretion of molecules, such as neurotransmitters, proteins, and SCFAs, directly affecting the immune system [112]. Probiotics are believed to exert anxiolytic-like effects through vagal pathways affecting areas like the periaqueductal gray and the central nucleus of the amygdala [113].

Regarding MDD, there is growing interest in whether probiotic-based therapies can ameliorate it, considering the comorbidity of depression with alterations in gut microbiota composition [114]. Recent studies have shown that probiotics positively affect individuals with pre-existing depressive symptoms, while their impact on mood symptoms in healthier populations is less significant [13,115]. Probiotics are known to change the sensitivity of the intestinal tract, regulate the stimulation threshold of intestinal neurons, and maintain the ecological stability of gut microbiota, thereby influencing the CNS and improving depression [116]. In a study by Kazemi et al., probiotic supplementation resulted in a decrease in the Beck Depression Inventory score compared to the placebo and prebiotic groups, highlighting the potential of this treatment in reducing MDD symptoms [117].

In exploring the effects of synbiotic treatments on psychiatric disorders, a notable randomized multicenter trial by Ghrobani et al. examined the adjunctive use of fluoxetine for MDD. In this study, 40 patients with a Hamilton Rating Scale for Depression (HAMD), with scores ranging from 17 to 23, initially received fluoxetine for 4 weeks. Subsequently, they were either administered a synbiotic capsule (alongside fluoxetine) or a placebo (with fluoxetine) for an additional 6 weeks. The findings revealed a more significant reduction in HAMD scores among patients treated with synbiotics compared to the placebo group, highlighting the efficacy of synbiotics as an adjuvant therapy in moderate depression [118].

Additionally, Zhang et al. conducted a meta-analysis involving 13 randomized controlled trials with 786 participants to evaluate the effects of prebiotics, probiotics, and synbiotics on patients with depression. The results showed that patients receiving these treatments experienced significantly improved depressive symptoms compared to those in the placebo group [119].

Intriguingly, studies involving fecal microbiota transplants from psychiatric patients to germ-free rodents have demonstrated the induction of symptoms similar to those of the donor’s disorders [120,121]. Moreover, certain probiotics or fecal microbiota transplants from healthy donors have been effective in alleviating symptoms and inducing positive outcomes in patients with psychiatric disorders [106,122].

These findings collectively suggest that the gut microbiome and the GBA play crucial roles in the onset and modulation of psychiatric disorders. Various related therapies, as shown in Table 2 of the relevant literature, have demonstrated promising results. However, it is essential to conduct further research to fully understand the benefits of prebiotic, probiotic, and synbiotic treatments. Additionally, a longer-term follow-up with individuals is necessary before these therapies can be widely implemented in clinical practice.

**Table 2 ijms-25-03340-t002:** The impact of therapies based on prebiotics and probiotics on mental disorders; FOS (Fructooligosaccharides); MDD (Major Depressive Disorder).

Author and Year	Objective	Outcomes on GM	References
Xu et al., 2022	The effect of probiotics on MDD.	Administration of *Lactobacillus rhamnosus zz-1* resulted in increased levels of the *Lachnospiraceae NK4A136 group*, *Bacteroides*, and *Muribaculum*.	[123]
Dandekar et al., 2022	The effect of probiotics on MDD.	The administration of a multi-strain probiotic formulation (Cognisol) was associated with increased ratios of *Firmicutes* to *Bacteroidetes*.	[124]
Mysonhimer et al., 2023	Fructooligosaccharides and Galactooligosaccharides as biomarkers for stress and inflammation.	The intervention did not alter biological markers of stress and inflammation; however, it resulted in an increase in *Bifidobacterium* levels.	[125]
Jiang et al., 2023	The influence of Fructooligosaccharides and Galactooligosaccharides prebiotics on gastrointestinal and blood-brain barrier dysfunction associated with stress-induced anxiety and depression.	Female mice exhibited greater susceptibility to stress and the effects of prebiotics compared to male mice.	[126]
Tian et al., 2021	The role of Butylated Starch in the context of Chronic Stress.	Administration of SCFA-acylated starches resulted in elevated levels of *Odoribacter* and *Oscillibacter*.	[127]

## 4. Interactions between Gut Microbiota and Psychiatric Medication

Over time, research has revealed that the administration of psychotropic drugs induces specific changes in the microbiome diversity. Such alterations have been observed in rodent models, where the administration of these drugs led to distinct modifications in the gut microbiota [128,129,130]. Moreover, studies have identified the antimicrobial properties of antidepressants and antipsychotics, demonstrating their ability to disrupt cell membrane integrity, mitochondrial activity, and key virulence factors of *Candida* spp., such as hyphae formation and the activity of SAP and phospholipase enzymes [131]. This body of research collectively contributes to the understanding of how psychiatric treatment may influence the brain–gut-microbiome (BGM) axis, with multiple studies supporting the hypothesis of such an interaction.

### 4.1. Antipsychotics

Similar to antibiotics, numerous medications, including antipsychotic drugs (APs), are known to alter the composition of gut microbiota by selectively eliminating certain bacterial species. Studies have documented this effect, emphasizing the impact of APs on microbial diversity [128,132]. Chlorpromazine, the first developed AP drug, was identified years ago as having antibiotic properties [133]. Its antibiotic efficacy, found to be more pronounced against Gram-positive bacteria compared to Gram-negative species, is attributed to alterations in the architecture and permeability of microbial membranes [134].

Research by Nehme et al. revealed that phenothiazines like chlorpromazine and thioridazine, as well as the thioxanthine flupenthixol, possess notable antibiotic effects. In contrast, newer generations of AP drugs such as clozapine, risperidone, olanzapine, and aripiprazole demonstrated limited antimicrobial activity [135]. A comprehensive in vitro study assessing over 1000 non-antibiotic drugs found that nearly all AP subclasses exhibited some degree of antibiotic activity, with about 24% of the non-antibiotics tested having detrimental effects on at least one bacterial strain. APs, along with chemotherapeutic agents and antihypertensives, were identified as the most potent antibacterials among the drugs examined [136].

A study by Maier et al. showed significant reductions in *Akkermansia muciniphila* populations in patients taking second-generation AP drugs, which are of particular concern due to their metabolic side effects [136]. Cussotto et al.’s research on rats indicated that olanzapine administration not only led to weight gain but also altered the gut microbiota, increasing *Firmicutes* and decreasing *Bacteroidetes*. Interestingly, when the experiment was replicated with germ-free rats, olanzapine did not induce weight gain, suggesting the gut microbiota’s role in this adverse effect of APs [130]. However, a study involving adult schizophrenia patients treated with olanzapine did not find analogous results in human fecal microbiota, suggesting a more complex interaction in humans [137].

Considering the long-term daily usage of APs, their antibacterial properties may significantly contribute to the growing issue of antimicrobial resistance, a phenomenon also observed with selective serotonin reuptake inhibitors (SSRIs) [138].

### 4.2. Antidepressants

Recent research indicates that gut microbiota may undergo alterations during major depressive episodes or in response to antidepressant treatments, which are potential confounding factors often not adequately considered [130,139,140]. A study by Chait et al. tested antidepressants such as phenelzine, desipramine, venlafaxine, bupropion, aripiprazole, and *(S) citalopram* on isolated commensal bacteria. The findings revealed that these drugs could inhibit the growth of dominant human microbiota phyla, such as *Bifidobacterium animalis* and *Bacteroides* fragilis, suggesting an antimicrobial capacity in regard to antidepressants on beneficial gut microbes [141].

In the context of antidepressants, fluoxetine (a selective serotonin reuptake inhibitor or SSRI) was found to induce minor changes in the microbiota, decreasing Deferribacteres and completely inhibiting the growth of *Succinivibrio* and *Prevotella* caecal taxa [130]. Lyte and colleagues further examined fluoxetine’s impact on healthy male mice’s microbiota, revealing dysbiosis and a decrease in genera such as *Lactobacillus johnsonii* and *Bacteroidales* S24-7, which was associated with altered body mass regulation. This suggests that fluoxetine-induced microbial depletion might contribute to side effects like weight loss [142]. Another study noted a decrease in genera *Prevotella*, *Oscillospira*, and *Ruminococcus* in female rats exhibiting depressive-like behavior when treated with fluoxetine during pregnancy and lactation [143]. A more recent investigation assessed the effects of five different antidepressants (fluoxetine, escitalopram, venlafaxine, duloxetine, and desipramine) on the gut microbiota of BALB/c mice. An increase in microbial β-diversity was observed in all treated groups, with a depletion of genera such as *Ruminococcus* and *Adlercreutzia*. Interestingly, the supplementation of *Ruminococcus flavefaciens* appeared to attenuate behavioral symptoms in duloxetine-treated mice, indicating potential anti-antidepressant effects [58]. Furthermore, Zhang et al.’s study examined the effects of fluoxetine and amitriptyline on gut microbiota in a rat model of chronic unpredictable mild stress (CUMS)-induced depression. Fluoxetine was more effective than amitriptyline in altering the levels of *Firmicutes* and *Bacteroidetes*. Both antidepressants significantly increased *Porphyromonadaceae* abundance, but an increase in *Bacteroidaceae* was specifically associated with amitriptyline. The study demonstrated a direct impact of oral antidepressant administration on the gut microbiome, providing a foundation for understanding their therapeutic functions and contributions to overall host health [144].

Shen et al.’s research focused on examining GM variations in patients with MDD treated with escitalopram. The study found a significant difference in the *Firmicutes*/*Bacteroidetes* ratio among three groups, with the follow-up group showing a notably lower ratio compared to the others. Additionally, alpha diversity was significantly higher in the MDD group compared to other groups, but there was no significant difference in alpha diversity between the control and follow-up groups. These findings suggest that the intestinal flora of depressed patients may trend towards normalization under escitalopram treatment [145].

Liskiewicz and colleagues observed a substantial increase in fecal microbiota biodiversity, particularly in alpha diversity, in hospitalized patients experiencing depressive episodes. This change was noted despite no significant differences in taxa abundance after six weeks of hospitalization and treatment with 5–20 mg of escitalopram [129]. These observations underscore the potential role of the direct antimicrobial effects of psychotropics in their mechanism of action, possibly through the regulation of microbiota.

Furthermore, low levels of serotonin have been associated with the onset of depression [146]. Antidepressant use could therefore selectively affect microbial strains capable of producing this neurotransmitter, potentially exacerbating the compromised health system in depression. A recent hypothesis proposes that alterations in gut microbial composition may activate pro-inflammatory pathways, exacerbating depressive symptoms and implicating the immune system in this complex interplay [147].

The challenge of antibiotic resistance on a global scale underscores the need for the development of novel antibiotics or antimicrobials. In addition to new drug discovery, drug repositioning could be explored as an alternative approach (Figure 4). This chapter has highlighted the antimicrobial properties of antidepressants and antipsychotics, suggesting their potential utility in combating bacterial infections.

## 5. Conclusions

In wrapping up our exploration, we have gauged the significance of the gut microbiota during the events of several psychiatric disorders. We proved how studies from literature have broadened the limits, showing insightful results in alleviating MDD, psychosis or anxiety. Therefore, probiotic and prebiotic supplements deserve some attention when it comes to treating psychiatric conditions. Furthermore, some promising approaches considering the antimicrobial effects of psychotropic drugs may thwart the ruthless dynamics of psychosis and depression, also providing some helpful remedies. Thus, further research may fill in the existing gaps regarding treatment for psychiatric disorders, as it is imperative to mitigate their influence and promote the overall health.

## Figures and Tables

**Figure 1 ijms-25-03340-f001:**
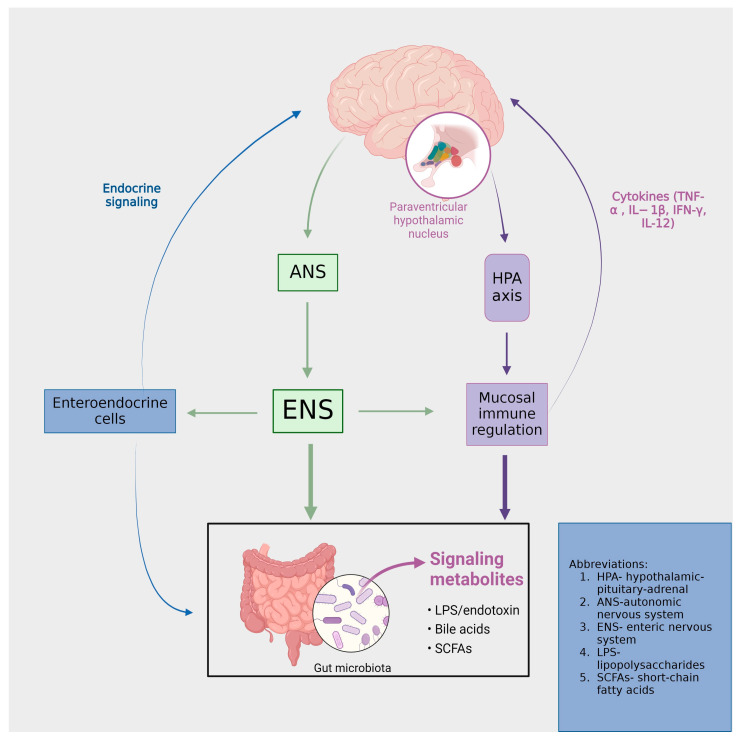
The functional and symptomatic ramifications of the brain–gut-microbiota axis are profound. Intestinal activities are subject to modulation by both the central nervous system (through the autonomic nervous system (ANS) and enteric nervous system (ENS)) and the intestinal microbiota. Signaling from microbial entities to the brain is facilitated via vagal and afferent neural pathways, as well as through the mediation of cytokines and neurotransmitters.

**Figure 2 ijms-25-03340-f002:**
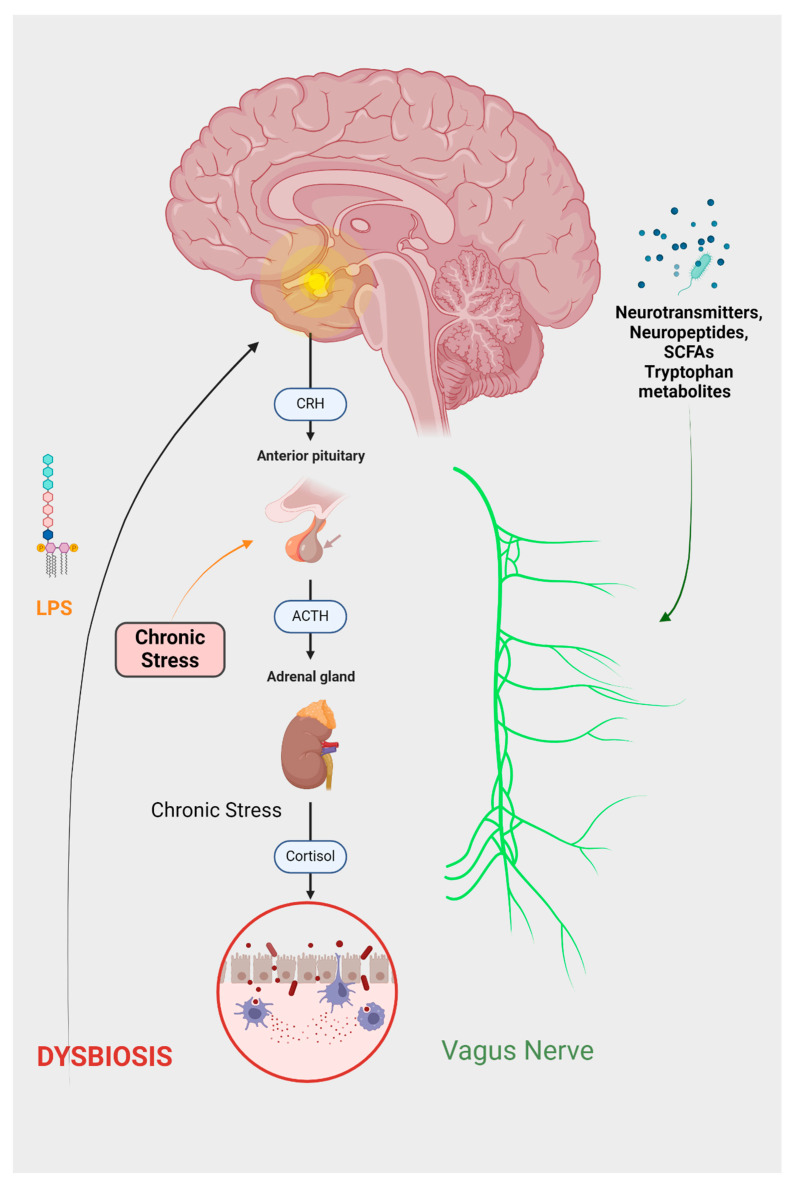
The depressive GBA can be summarized as follows: Chronic stress leads to the activation of the HPA axis, which increases cortisol production. Elevated cortisol levels disrupt the equilibrium of gut microflora, contributing to increased intestinal permeability, commonly referred to as ‘leaky gut’. Consequently, harmful substances, particularly LPS, can access the brain. Cortisol also initiates an inflammatory response and activates the endocannabinoid system. The altered gut microbiota produces various substances, including neuropeptides, hormones, and SCFAs. The effects of these substances, in conjunction with inflammatory mediators, are predominantly facilitated through the vagus nerve.

**Figure 3 ijms-25-03340-f003:**
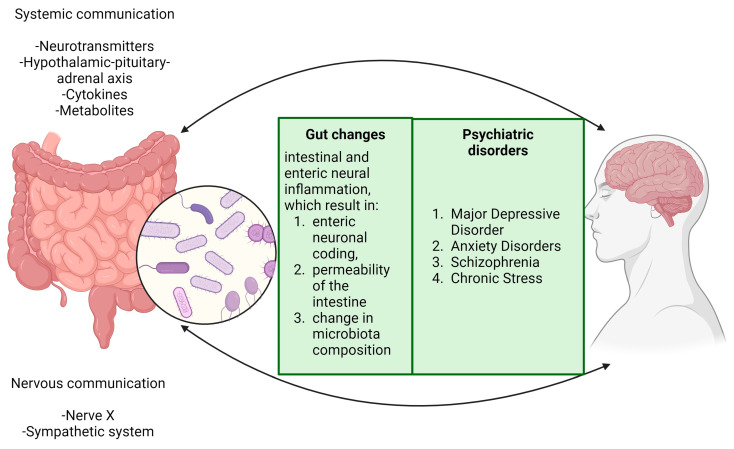
The brain and digestive system are interconnected in a way that dysbiosis in gut microbiota can initiate a series of biological responses. This imbalance in the gut microbiota activates the intestinal immune system, leading to increased intestinal permeability and bacterial translocation. These processes are key contributors to various neurological outcomes, including neuroinflammation, epigenetic changes, cerebrovascular alterations, accumulation of amyloid β, and aggregation of α-synuclein proteins. These modifications are associated with the development and progression of several neurological and psychiatric conditions, such as hypertension, Alzheimer’s disease, Parkinson’s disease, stroke, epilepsy, and autism. Moreover, specific nuclei in the brainstem, particularly the nucleus tractus solitarius and the dorsal motor nucleus of the vagus, are crucial in regulating gastric motor functions. They facilitate this regulation through bidirectional communication via the vagus nerve, highlighting the complex interaction between the brain and gastrointestinal system. This interaction is vital not only for maintaining physiological balance but also for responding to pathophysiological changes. This intricate relationship underscores the significance of the gut–brain axis throughout periods of both health and disease.

**Figure 4 ijms-25-03340-f004:**
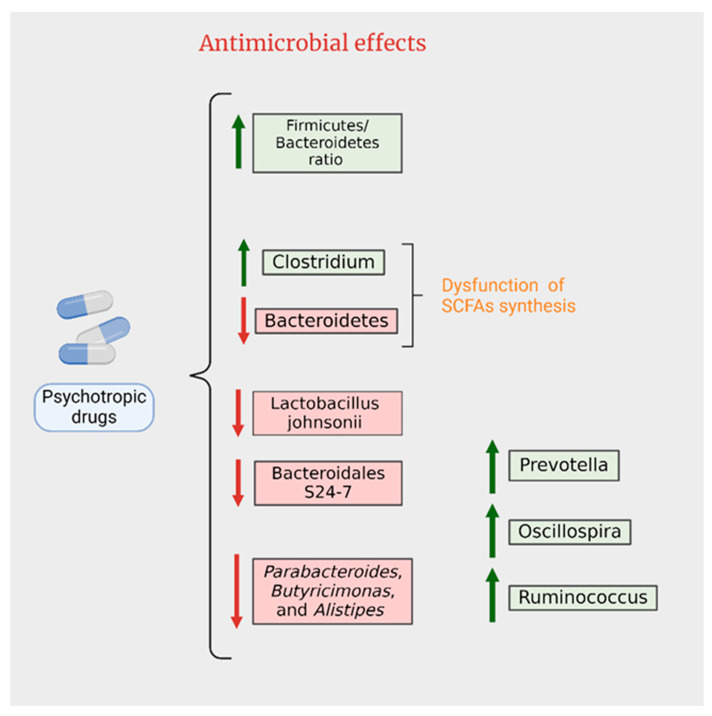
Antimicrobial effects of psychotropics.

**Table 1 ijms-25-03340-t001:** Studies depicting various correlations between gut microbiota (GM) and psychiatric disorders; PTSD = post-traumatic stress disorder; ADHD = Attention deficit/hyperactivity disorder; ASD = Autism Spectrum Disorder; MDD = Major Depressive disorder; HAMD = Hamilton Depression Rating Scale.

Author and Year	Psychiatric Disorder	GM Effects	References
Hemmings et al., 2017	PTSD	PTSD is associated with Decreased levels of *Actinomycetota*, *Lentisphaerae* and *Verrucomicrobiota*	[89]
Xu et al., 2017	Chronic stress	Increased prevalence of Prevotella and reduced levels of *Veillonella*, *Paraprevotella*, *Odoribacter*, and *Ruminococcus* in individuals experiencing chronic stress.	[90]
Stevens et al., 2019	ADHD	Elevated concentrations of *Actinomycetota* in patients with ADHD.	[91]
Wang et al., 2020	ASD	In patients with ASD, an increase in *Clostridium* and *Ruminococcus* was observed following interventions with probiotics and fructo-oligosaccharides; conversely, a reduction in *Bifidobacterium* longum was noted in ASD patients.	[92]
Chen et al., 2021	MDD	Levels of *Anaerotruncus*, *Anaeroglobus*, and *Parabacteroides* were found to correlate with Hamilton Depression Rating Scale (HAMD) scores.	[55]

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
