# Peer review of "Mind, Mood and Microbiota—Gut–Brain Axis in Psychiatric Disorders"

_ijms, 2024, doi:10.3390/ijms25063340_

Round 1

Reviewer 1 Report

Comments and Suggestions for Authors

The authors aimed to emphasize the intricate correlations between psychiatric disorders and gut microbiota, mentioning the promising approaches involving the modulation of probiotic and prebiotic treatments, as well as the antimicrobial effects of psychotropic medication. The manuscript is well-structured and well-written. I have some comments on this manuscript.

The relationship between the microbiota-gut-brain axis and health issues, including mental health, has been extensively discussed in previous publications, such as the study by Góralczyk-BiÅ„kowska et al. (2022) titled 'The Microbiota–Gut–Brain Axis in Psychiatric Disorders,' published in this journal. 

The novelty of the current manuscript lies in its expansion to include discussions on "Mind" and "Mood". However, these concepts, as titled by the authors, are not fully elucidated within the content. I hope the revised manuscript will be more comprehensive.

Please rewrite this sentence "Additionally, the effectiveness of mindfulness-based cognitive therapy in populations with high trait anxiety provides further evidence of gut-brain communication" which refers to the role of gut microbiota (line 323-324).

Minor comments:

Please correct some typos “promissing” (line 25, line 581);  “antimicriobial effects” (line 25); “we will one day pe able to treat” (line 80); “succeded in” (line 83); “whent” (line 580); “Neurotransmiters” (Figure 3)

Figure 1. Please provide explanations for abbreviations

Author Response

  • Comment: The novelty of the current manuscript lies in its expansion to include discussions on "Mind" and "Mood". However, these concepts, as titled by the authors, are not fully elucidated within the content. I hope the revised manuscript will be more comprehensive.

Response: Thank you for your feedback! In the revised version, we have tried to clarify more the "mind" and "body" aspect of the manuscript

  • Comment: Please rewrite this sentence "Additionally, the effectiveness of mindfulness-based cognitive therapy in populations with high trait anxiety provides further evidence of gut-brain communication" which refers to the role of gut microbiota (line 323-324).

Response: Thank you for your suggestion. We have revised the sentence.

  • Comment: Please correct some typos “promissing” (line 25, line 581);  “antimicriobial effects” (line 25); “we will one day pe able to treat” (line 80); “succeded in” (line 83); “whent” (line 580); “Neurotransmiters” (Figure 3)

Response: Thank you for bringing these errors to our attention. We made sure that all the identified typos are corrected in the revised version of the manuscript.

  • Comment: Figure 1. Please provide explanations for abbreviations

Response: We have added explanations for all abbreviations used in Figure 1.

Reviewer 2 Report

Comments and Suggestions for Authors

In the manuscript submitted to me for review entitled "Mind, mood and microbiota- Gut-Brain axis in psychiatric disordersthe authors Corneliu Toader, Nicolaie Dobrin, Daniel Costea, Luca Andrei Glavan, Razvan-Adrian Covache-Busuioc, David-Ioan Dumitrascu, Bogdan-Gabriel Bratu, Horia Petre Costin and Alexandru Vlad Ciurea present an extensive review examining the relationship between psychiatric disorders and gut microbiota.

The systematization of knowledge on the problem under consideration can contribute to the further expansion of research on the gut microbiota and its impact on human health, including the impact on brain processes through the brain-gut axis. This may contribute to the development of better probiotics and prebiotics that positively affect human mental health.

In their study, the authors used 147 references, of which 2/3 were from the last 5 years. This shows that the topic addressed in the present manuscript is of great interest to many research teams and would be of interest to IJMS readers as well. The included data are summarized using 4 well-designed and presented figures and 2 tables. The conclusions drawn correspond to the data presented in the manuscript.

My remarks and recommendations to the authors are:

1.     I recommend that authors review the „Instructions for Authors“, and in particular the way references are cited in the text. Basically they presented them the right way, but they used separate brackets for each reference (where there are more than one).

For example: [1], [2], [3]

but this is how several brackets are accumulated in one place. A more appropriate way is to present all the references in one bracket: [1, 2, 3].

2. On line 123, the abbreviation GABA is entered, the full name of which has not been presented until now. Let it be added.

3. A „Conflict of interest“ section should be added to the Back Matter of the manuscript, where the authors declare that they have no conflict of interest (see „Instructions for Authors“).

4. Bearing in mind that this is a review article that is supposed to attract the attention of readers and will most likely receive good reviews and be cited by a significant number of other authors, in my opinion everything should be presented as fully as possible. In the „References“ section, there are some references in which not all authors are represented (for example, in numbers 3, 7, 11, 17, 19, 24, 25, 27, 30, 37, 38, 44, 45, 52, 53, 5, 55, 58, 59, 60, 64, 6, 66, 67, 68, 71, 73, 75, 76, 78, 79, 83, 84, 86, 87, 88, 89, 90, 92, 94, 96, 98, 101, 105, 108, 109, 110, 111, 119, 120, 124, 126, 127, 128, 129, 130, 134, 135, 136, 137, 138, 140 and 146). Let all the authors be added. In my opinion, this would be of great benefit to the readers of the manuscript.

5. In the „References“ section, reference number 4 does not indicate the year of publication. Let it be added.

Author Response

  • Comment:  I recommend that authors review the „Instructions for Authors“, and in particular the way references are cited in the text. Basically they presented them the right way, but they used separate brackets for each reference (where there are more than one).

Response: We ensured that references are cited in a consistent manner according to the guidelines provided, including instances where multiple references are cited within the same bracket.

  • Comment: On line 123, the abbreviation GABA is entered, the full name of which has not been presented until now. Let it be added. 

Response: We have addressed that in the newly submitted manuscript.

  • Comment: A „Conflict of interest“ section should be added to the Back Matter of the manuscript, where the authors declare that they have no conflict of interest (see „Instructions for Authors“).

Response: We have addressed that in the newly submitted manuscript

  • Comment: Bearing in mind that this is a review article that is supposed to attract the attention of readers and will most likely receive good reviews and be cited by a significant number of other authors, in my opinion everything should be presented as fully as possible. In the „References“ section, there are some references in which not all authors are represented (for example, in numbers 3, 7, 11, 17, 19, 24, 25, 27, 30, 37, 38, 44, 45, 52, 53, 5, 55, 58, 59, 60, 64, 6, 66, 67, 68, 71, 73, 75, 76, 78, 79, 83, 84, 86, 87, 88, 89, 90, 92, 94, 96, 98, 101, 105, 108, 109, 110, 111, 119, 120, 124, 126, 127, 128, 129, 130, 134, 135, 136, 137, 138, 140 and 146). Let all the authors be added. In my opinion, this would be of great benefit to the readers of the manuscript.

Response: All authors have been appropriately listed for better transparency and acknowledgment in the revised manuscript.

  • Comment: In the „References“ section, reference number 4 does not indicate the year of publication. Let it be added. 

Response: We have addressed that in the newly submitted manuscript. 

in the end, we would like to thank you for your positive feedback regarding our manuscript. Your expertise has helped us improve the quality our work.

Round 2

Reviewer 1 Report

Comments and Suggestions for Authors

Thank you for your effort in revising the manuscript. The mentioned typos have been corrected. However, upon review, I noted that the revised manuscript did not substantially differ from the previous version, except for the addition of a paragraph:

"In the complex realm of human cognition and physiology, the interactions between mind, mood, and the brain-gut microbiota arise as an intriguing frontier. This suggests that the gut microbiota evolves alongside the gut-brain, brain, and consequently, the mind. Moreover, the microbiota play an insightful role in both the maturation and operation of the brain, suggesting their influence in the complex orchestra that regulates our mood [10,11]. Major Depressive Disorder (MDD), Anxiety, and Bipolar Disorder are psychopathologies primarily affecting the mind, categorized as mind disorders, influenced by alterations of BGM." (lines 75-81).

Therefore, as I mentioned previously, the novelty of the current manuscript seems limited because of the overlap of ideas with a previously published article: Góralczyk-BiÅ„kowska et al. (2022). The Microbiota–Gut–Brain Axis in Psychiatric Disorders.